# Immunogenicity and safety of measles-mumps-rubella vaccine delivered by the aerosol, intradermal and intramuscular routes in previously vaccinated young adults: a randomized controlled trial protocol

**Sumanta Saha**[1]*, **Melanie Millier**[1], **Ari Samaranayaka**[1], **Liza Edmonds**[2], **Emma Best**[3], **James Ussher**[1], **Andrew Anglemyer**[1], **Jennifer Lee**[1], **Michael Tatley**[1], **Felicity Cutts**[4], **Rob van Binnendijk**[5], **Peter McIntyre**[1]*

1 University of Otago, Otago New Zealand, 2 Victoria University, Wellington New Zealand, 3 The University of Auckland, Auckland New Zealand, 4 The London School of Hygiene & Tropical Medicine, Bloomsbury, UK, 5 National Institute for Public Health and the Environment, Bilthoven The Netherlands

* peter.mcintyre@otago.ac.nz (PM); sumanta.saha@uq.net.au (SS)

## Abstract

### Background

There are increasing reports of outbreaks of measles in countries that achieved measles elimination using two doses of measles-mumps-rubella (MMR) vaccine, particularly in health care settings. While responses to a third dose of MMR in two-dose recipients have been examined, these studies have all administered MMR by the standard (intramuscular or subcutaneous) route, and data on the duration of antibody are limited. We have developed a protocol for an open-label parallel-arm randomized-controlled trial to compare measles antibody responses and safety after intradermal and aerosol administration of MMR with intramuscular, the usual mode of administration in Aotearoa (New Zealand).

### Methods

Eligible participants are aged ≥18 years who have previously received two doses of the MMR vaccine and based on levels of IgG antibody to measles or mumps below the threshold for seropositivity in commercially available screening tests are required to receive the MMR vaccine prior to entering health professional training programs at Aotearoa universities. The participants will be randomized to three routes of administration (1:1:1) to receive the MMR vaccine by the intradermal (via microneedle), intrapulmonary (via vibrating mesh nebulizer), or intramuscular routes. The primary objective is to determine the proportion of participants who attain levels of measles IgG antibody above the seroprotective threshold using a multiplex bead-based immunoassay, with those in the lowest quartile validated by plaque neutralization assay, at days 6–8, 13–15, 28–42, and at 12–18 months post-vaccination. Secondary objectives include a fold increase in the geometric mean concentration of IgG antibody from baseline, and systemic and local

**Data availability statement:** No datasets were generated or analysed during the current study.

**Funding:** The trial has received funding support from Health Research Council, New Zealand. Prof. Peter McIntyre has received funding support from Health Research Council, New Zealand for the trial (https://www.hrc.govt.nz/resources/research-repository/measuring-and-boostingwaning-immunity-measles-young-adults). Dr Sumanta Saha was funded by Te Niwha (Funding reference no.TN/PLSP23/46/OUPM), the Infectious Diseases Research Platform – co-hosted by Environmental Science and Research (ESR) and the University of Otago and provisioned by the Ministry of Business, Innovation and Employment, New Zealand. The funders had no role in study design, data collection and analysis, decision to publish, or preparation of the manuscript.

**Competing interests:** The authors have declared that no competing interests exist.

reactions following delivery of MMR by each method. The trial is registered with the Australian New Zealand Clinical Trials Registry (ANZCTR; https://www.anzctr.org.au/Default.aspx; trial registration no. ACTRN12623000130662).

## Introduction

Measles is a highly contagious infectious disease caused by the measles virus, a negative-sense single-stranded RNA virus of the Morbillivirus genus and Paramyxoviridae family. [1] Measles has an incubation period of 10–14 days. On entering the epithelial cells of the upper airway of a susceptible host, the virus first infects the myeloid cells (e.g., macrophages and dendritic cells),[2, 3] then migrating to infect regional lymphoid cells and T- and B-cells, which eventually enter the bloodstream and infect the epithelial cells of the skin and the submucosa of the respiratory tract. The latter results in coryzal symptoms and cough with spread of infectious viral particles via respiratory droplets and small aerosols to other individuals. [4, 5] Measles causes a severe systemic disease clinically characterized by symptoms including fever, cough, coryza, conjunctivitis, and rash, but in high income settings rarely mortality. [1] Live attenuated measles vaccine in combination with rubella (measles-rubella, MR) or with both rubella and mumps (measles-mumps-rubella, MMR) is highly effective in protecting against all targeted diseases.[1] From 2000, the steadily increasing coverage of the measles vaccine globally led to dramatic reductions in measles infections and deaths in children, and the achievement of measles elimination in the World Health Organization (WHO) Region of the Americas which in 2016 led to the World Health Assembly setting the target of measles elimination in five of the six WHO regions by 2020. However, by the end of 2022, although 83 countries (43% of countries worldwide) had achieved measles elimination at some point, no WHO region had sustained measles elimination status, with a substantial global resurgence in 2019.[6, 7]

Measles remains endemic in many countries. As a result, countries where elimination has been achieved following prolonged periods of high two-dose measles vaccination coverage accompanied by very low measles incidence, outbreaks occur due to imported cases. In these outbreaks, cases of measles infection in people with two documented measles vaccine doses have been increasingly documented. Examples include outbreaks where 26/232 (11%) cases in California (2000–2015),[8] 6/32 (19%) cases in New South Wales, Australia in 2012,[9] and 11/94 (11.7%) cases in South Korea in 2019, [10] occurred in persons with two previous MMR doses. Similarly, in a large outbreak of over 2,100 cases in New Zealand in 2019, about 10% of cases in young adults occurred in two-dose MMR recipients.[11] This has raised the question of whether such countries need to consider a third dose of measles-containing vaccine to sustain elimination in the presence of increased and persistent transmission globally and the threat of outbreaks. Although secondary measles vaccine failures are usually less clinically severe and contagious, transmission is well reported in the context of outbreaks in countries which have achieved measles elimination, with two-dose vaccinated cases accounting for 9-19% of the total adult cases [7,9] and potentially contributing to transmission and failure to maintain elimination status. Data on the magnitude and persistence of antibody responses after a third dose of measles-containing vaccine in adults with measles antibody below protective levels are sparse, with existing data primarily from seropositive adults. [12, 13] Should a third dose of MMR be recommended in adolescents or young adults, uptake is likely to be improved by non-needle routes of administration, such as intradermal and aerosol delivery. However, data on safety and immunogenicity are lacking both for alternate routes of administration and among adults who are seronegative

despite documented prior vaccination. Aerosol vaccine delivery, pioneered in Mexico using the "Classical Mexican Device," a handmade nebulizer driven by a battery-operated compressor, was extensively used in the 1980s for mass childhood immunization against measles.[14, 15] Aerosol delivery of measles vaccine was revisited in a WHO-sponsored trial in Indian infants, which selected the Aerogen Solo vibrating mesh nebulizer (VMN) for use as it had been shown to reliably deliver aerosolized measles vaccine to the lower respiratory tract. [15] In a small study of 27 young adults in Mexico, delivery of MMR vaccine using a similar VMN device was found to have an acceptable safety and immunogenicity profile. [16] More recently, during the COVID-19 pandemic, aerosol delivery of an adenoviral vector vaccine against SARS-CoV-2 (Cansino) was extensively used in China, with a favorable safety and immunogenicity profile.[17,18] VMNs generate 3–5 μm size particles that deposit the vaccine deep into the lung tissue when inhaled.[19] Other than the study cited above,[16] previous studies in adults on the safety and immunogenicity of measles-containing vaccines (MCV) by route of administration have compared subcutaneous administration with aerosol delivery using the Classical Mexican Device for aerosol generation, used vaccine strains not used in high-income elimination settings (such as Edmonston Zagreb (measles) and Leningrad Zagreb or Rubini (mumps) strains, and have predominantly included seropositive subjects.[20, 21] No previous study has evaluated the safety and immunogenicity of MMR vaccines using the Schwarz strain delivered by VMN to seronegative adults who have previously received two measles-containing vaccine doses in childhood. Likewise, although intradermal vaccine delivery has been studied for vaccines against influenza,[22] polio[23] and hepatitis B,[24] responses to intradermal MCV have not been examined in seronegative adults.

Given evidence that measles cases have been increasingly recognised in measles elimination settings in two-dose vaccinated young adults in childhood, whether the strength and/or duration of protection can be improved is a key question. In this unblinded safety and immunogenicity study of a third dose of MMR vaccine, we will compare aerosol and intradermal delivery of MMR vaccine with the intramuscular route routinely used in New Zealand in seronegative adults in an open-label parallel three-arm randomized-controlled trial. We aim to test the hypotheses that: 1) a higher measles-specific immune response will be induced by aerosol delivery, with similar or superior local and systemic safety profile to intramuscular delivery; 2) a higher immune response will be induced by intradermal delivery, with similar or superior local and systemic safety profile than intramuscular delivery.

## Methods

This protocol is presented following the Standard Protocol Items: Recommendations for Interventional Trials (SPIRIT) 2013 guidelines (SPIRIT checklist: Table in S1 File).[25]

### Study setting

We are conducting a multicenter three parallel-arm randomized-controlled open-label trial to compare immunogenicity across aerosol, intradermal, and intramuscular MCV deliveries (as MMR vaccine) at designated trial sites in Aotearoa. At tertiary institutions where serological screening for measles and mumps antibodies is required, we commenced enrolling participants in 2023 and are continuing to enroll in 2024 and 2025. The trial is registered at the Australian New Zealand Clinical Trials Registry (ANZCTR; https://www.anzctr.org.au/Default.aspx; trial registration no. ACTRN12623000130662; trial acronym: MAXXED: Measles vaxx routes of delivery). The trial started recruiting in February 2023 and will end in December 2025.

## Ethics statement

This study will be conducted in compliance with the Declaration of Helsinki and its amendments and the applicable regulations within Aotearoa. The trial received approval (Ethics Approval number: 13681) from the Northern A Health and Disability Ethics Committees (Ministry of Health, Health and Disability Ethics Committees, Wellington, New Zealand) on 10/01/2023 (S5 File).

Given the significant impact of the measles outbreak in Aotearoa in 2019 for whānau Māori (indigenous peoples of Aotearoa), opportunities arise in the area of optimizing vaccination in Māori communities. Optimizing Māori student representation among the trial participants is important given the opportunities for non-needle-based and/or less invasive measles vaccination methods which might provide for measles protection among Māori people by increasing vaccine uptake. A senior Māori researcher has been part of this trial to support Māori involvement in the study. This is in addition to the Tauiwi (non-Māori) researchers as part of pro-equity approach and commitment to improved health outcomes for whānau Māori.

## Eligibility criteria

**Inclusion criteria.**  Eligible trial participants are 18 to 49 years of age who have received two doses of MMR vaccine and have measles IgG antibody levels which are negative or equivocal on commercial assays (for example ≤ 16.5 arbitrary units (AU) on the DiaSorin assay [26]), are capable and willing to provide written informed consent, complete an electronic diary post-vaccination (requiring access to an internet-enabled device), and participate in all required study visits. Previous receipt of two MMR doses will be verified by written records with dates of administration, or if unavailable, a participant signed declaration that two doses have been received will be accepted. Students who are eligible to participate based on their serological results, but lack acceptable evidence of two prior MMR doses, are not excluded from enrollment and will be separately evaluated. Tertiary education institutions in Aotearoa enrolling students in health professional courses typically require serological confirmation of detectable measles and mumps IgG antibodies in an approved commercial assay, with additional evidence of receipt of two MMR doses prior to commencing clinical placements. At institutions participating in the study, student health service staff contact students with antibody below the assay cut-off for measles or mumps to have an MMR dose, with subsequent repeat serological testing, irrespective of previous MMR vaccination history.

*Exclusion criteria:*

1. History of acute illness during the five days before study vaccine administration. Acute illness is defined as symptoms requiring antipyretic medication and/or absence from educational attendance, regardless if any diagnostic test for infection has been performed.

2. Contraindications to MMR as listed in the New Zealand Immunization Handbook[27]:

   a) Known anaphylactic reaction following previous MMR administration.

   b) Severe immunocompromise defined as impaired cell-mediated immunity, including untreated malignancy, type 1 interferon receptor signaling pathway defect, immunosuppressive drug therapy (including high-dose steroids), high-dose radiotherapy, human immunodeficiency virus infection with severely impaired T cell immunity.

c) Receipt of another live vaccine within the previous four weeks. Simultaneous administration is not precluded.

d) Pregnancy. Female participants not known to be pregnant at the time of enrollment but who subsequently find that they are pregnant, may continue to attend follow up visits required for the trial if they wish.

e) Intravenous immunoglobulin or blood transfusion during the preceding 11 months.

### Recruitment, randomization, and intervention allocation

Study data will be collected and managed using REDCap [28] electronic data capture tools hosted and supported by the University of Otago. Students requiring MMR vaccine, and registering interest in participating in the trial, are contacted by research team personnel for more detailed discussion of trial participation. Eligible students who verbally agree to participate will electronically sign a provisional consent form accessed through a secure database link (page 7 in S2 File) and then automatically get randomized to an intervention group by the randomization algorithm uploaded to the REDCap randomization module. Participants will sign a paper consent form (S3 File) at the beginning of the in-person vaccination appointment (study visit 1) and are then advised of their allocated intervention. Trial personnel and participants are aware of interventions allocated (open-label). Fig 1 shows the participant outline depicting trial events.

| TIMEPOINT | Enrolment | Allocation | STUDY PERIOD Post-allocation | | | | | | | | |
|---|---|---|---|---|---|---|---|---|---|---|---|
| | | | $t_1$ | $t_2$ | $t_3$ | $t_4$ | $t_5$ | $t_6$ | $t_7$ | $t_8$ | $t_9$ |
| | $-t_1$ | 0 | Intervention visit | 12 hr | 24 hr | 48 hr | 3-5 days (follow up visit) | 6-8 days (follow up visit) | 13-15 days (follow up visit) | 28-42 days (follow up visit) | 12 months (follow up visit) |
| **ENROLMENT:** | | | | | | | | | | | |
| *Eligibility screening[a]* | X | | | | | | | | | | |
| *Allocation* | | X | | | | | | | | | |
| *Informed consent* | | | X[b] | | | | | | | | |
| **INTERVENTIONS:** | | | | | | | | | | | |
| *Intramuscular MMR* | | | X | | | | | | | | |
| *Intradermal MMR* | | | X | | | | | | | | |
| *Aerosol MMR* | | | X | | | | | | | | |
| **ASSESSMENTS:** | | | | | | | | | | | |
| *Active adverse effect monitoring (e-diary)* | | | | X | X | X | | | | | |
| *Staff enquiry about side effects at visits* | | | | | | | X | X | X | | |
| *Oral fluid collection* | | | | | | | X | X | X | | |
| *Post-vaccination blood collection* | | | | | | | | X | X | X | X |

[a] participant with antibody levels below threshold to measles component
[b] pre-intervention written consent obtained on the day of intervention

**Fig 1. Trial procedure schedule.**

## Interventions

**Aerosol delivery.** The Aerogen Solo (S4 File) is pre-assembled into the Aerogen Ultra handheld device (Aerogen Ltd, Galway, Ireland) and individually packaged for use (one device per aerosol-arm participant) in the trial. When powered, the device generates aerosol particles by pumping liquid from the medication cup through the vibrating mesh aperture plate using the piezoelectric effect.[19,29] Immediately prior to vaccine administration, the device will be 'primed' with 2–4 drops of sterile saline and the participant will inhale the aerosolized saline via the mouthpiece to familiarize themselves with the administration method and to prevent the vaccine from adhering to the sides of the chamber. The freeze-dried PRIORIX MMR pellet is reconstituted to a total volume of 0.3 mL by adding manufacturer provided diluent, and injected into the medication cup of the Aerogen Solo once saline priming has been completed. Participants are advised to rest for about 10 minutes, during which they can read and sign the consent form and resolve any queries about the trial with the research team prior to aerosol administration. Participants insert the mouthpiece and, when the device is powered on, inhale the MMR dose by breathing naturally via the mouthpiece for 30–60 seconds until all visible reconstituted vaccine in the medication cup has been converted to aerosol and all mist in the reservoir has been inhaled under supervision of the vaccinator.

**Intradermal delivery.** Intradermal administration of vaccine uses the Micronjet600™ device (NanoPass Technologies Ltd, Israel; S4 File), a device consisting of three 600 μm long microneedles (https://www.nanopass.com/). PRIORIX MMR vaccine is reconstituted to a total volume of 0.3 mL of sterile water and drawn up into a 2mL syringe. After attachment to the syringe, the microneedle is inserted into the participant's skin in the deltoid region at a 45-degree angle in accordance with manufacturer instructions, and the vaccine dose is manually delivered to the intradermal region. A visible bleb indicates successful intradermal delivery.[30] The Micronjet600™ device has received usage approval in the US and the European Union.[31]

**Intramuscular delivery.** After reconstituting the vaccine in 0.5 ml of diluent, the vaccine will be injected at a 90-degree angle in the outer part of the upper arm using a 23–25-gauge needle to deliver the vaccine deep into the deltoid muscle.

**Vaccine.** PRIORIX MMR vaccine (GSK, USA), a lyophilized vaccine preparation containing a mixture of live attenuated strains of measles (Schwarz strain), mumps (RIT 4385, derived from Jeryl Lynn strain), and rubella (Wistar RA 27/3 strains) viruses propagated in chick embryo tissue cultures (mumps and measles) or MRC5 human diploid cells (rubella), used for the national immunization program in Aotearoa, is used for all participants.(31,32) Though using the same vaccine lot would be ideal, it would not be feasible as our study will be conducted at different geographic locations over a prolonged period (up to 3 years). However, the vaccine lots will be recorded, and antibody responses by vaccine lot will be examined. The vaccination records of each participant will contain the lot number and expiration date of the vaccine administered.

**Aerosol exposure to study personnel and household contacts.** The risk of exposure of vaccinators or other personnel to aerosolized vaccine virus is very low, as the mesh separates the breathing circuit from the medication chamber.[32–34] A study in Mexico did not find any evidence of vaccine virus transmission from aerosol vaccine recipients to study staff or household members.[32]

**Cross-contamination.** Despite the low leak potential of the aerosol delivery method, cross-contamination-preventing steps will be implemented, including vaccination in well-ventilated rooms and using separate rooms for aerosol and other routes of administration, with at least a 5-minute gap between vaccinations.

**Intervention providers.** All trial participants will be vaccinated by trained vaccinators equipped for vaccination by the intradermal and aerosol route through familiarity with manufacturer instructions and training videos for the specific devices used.

## Specimens and outcomes

**Venous blood collection.** The venous blood collection at baseline and 6–8 days, 13–15 days, 28–42 days, and 12–months post-vaccination will collect whole blood in serum separation tubes. Serum will be isolated following centrifugation of coagulated whole blood (1,800 x g for 10 min at 4ºC) and frozen aliquots of all samples will be sent to the National Institute for Public Health and the Environment (RIVM) in the Netherlands for testing to determine the serum antibody levels against measles virus using RIVM's bead-based multiplex immunoassay (MIA).[26] MIA is well-correlated with plaque reduction assays, the gold standard for antibody detection.[26] An anti-measles IgG titer of ≥ 120 mIU/ml will be considered seroprotective.[35] The correlation of the IgG values by MIA with the gold standard will be substantiated on the sera of subjects whose MIA IgG level is in the lowest quartile, which will be re-tested by the plaque reduction neutralization test.

**Oral fluid collection.** During the first three follow-up visits within two weeks post-vaccination (Fig 1) oral fluid collection will be carried out by the participant via self-swabbing the gingiva for one minute using the S14 Oracol Plus Saliva Collection Device (Malvern Medical Developments, UK).[36] Post-collection, swabs will be temporarily stored chilled prior to centrifugation (4ºC for 10 min at 1,800 x g) on the same day as the specimen is obtained, with longer-term post-centrifugation storage of oral fluids at −80ºC. Oral fluid will not be collected if there is any gum bleeding and participants will be advised to avoid eating or drinking during the 30 minutes prior to the study visit. Oral fluid specimens will be tested by reverse transcription polymerase chain reaction (RT-PCR) to specifically identify the presence of measles vaccine viral sequence.[26,37–39]

## Data collection

The participant data collection forms will record the following information (S2 File): demographic details, including country of birth (along with ethnicity) and countries where a participant has lived, previous measles infection, and vaccination history; smoking and vaping details; past illness history; current medication and dietary supplement usage; and allergy history. Self-reported post-vaccination adverse effects data will be gathered from the participants between 12–48 hours post-vaccination using e-diaries (participants are prompted to enter data in a survey format accessed through a secure REDCap database link automatically sent via SMS after 0700 hours at approximately 12, 24 and 48 hours post-vaccination) and solicited adverse effects data at 3–5, 6–8, and 13–15 days post-vaccination by a research team member during their first three in-person follow up visits (S2 File).

## Retention

To acknowledge participant time and potential discomfort related to the study, participants will receive a $125 (New Zealand dollars) voucher for their donated time and effort. Some participants may require other vaccines under their institutions' vaccination policy (e.g., hepatitis B, varicella vaccines).

## Participant withdrawal

Participants can withdraw from the study at any stage. The research team will continue to attempt to contact participants during the trial period unless consent is withdrawn.

## Data management and monitoring

All participant data is captured and stored via the REDCap platform. De-identified datasets will be prepared for analysis to ensure data security and protection of participant identity. Only necessary data for specimen identification will be sent to the testing laboratory (RIVM), and the principal investigator and designated members will monitor data entry at three monthly intervals and decide on any amendments needed (safety monitoring stated below).

## Safety

All participants will be observed for 30 minutes post-vaccination for symptoms of anaphylaxis or severe reaction like increased heart rate, breathing difficulty, feeling of faintness, or tingling sensation of mouth. Solicited adverse localized injection site reactions for intramuscular and intradermal recipients include pain, swelling and redness (with an option to upload photographs to REDCap available). Solicited adverse systemic reactions for all participants will include fever, vomiting, diarrhea, headaches, fatigue/tiredness, joint and muscle pain, and rashes (also with option to upload photographs to REDCap) (S2 File). Respiratory symptoms like cough, sore throat, runny nose, breathing difficulty, and chest discomfort will be sought. Redness and swelling are graded as absent, mild (grade 1; > 2–5 cm), moderate (grade 2; > 5–10 cm), or severe (grade 3; > 10 cm). Participants who record a grade 3 reaction are asked to contact the study team by telephone to determine if an onsite visit is required. The study team will also contact participants about any systemic symptom which is not resolving. Local and systemic reactions meeting criteria for grade 4 must be graded by a physician, and all local or systemic grade 4 events will be reported to the New Zealand Pharmacovigilance Centre for Adverse Reaction Monitoring (CARM), Aotearoa. The principal investigator or the sponsor or designee will categorize a side effect as serious if it leads to one of the following: a life-threatening event that puts a participant at immediate risk of death and hospitalization. During the study period, all serious adverse events will be reported to the study sponsor within 24 hours and to the CARM (https://pophealth.my.site.com/carmreportnz/s/). Such events will be independently assessed by the medical monitor assigned by the CARM.

## Statistical methods

**Sample size.** Over the two years 2023–2024, we anticipate ~ 1400 students will be screened for eligibility. When comparing immunogenicity of aerosol delivery and intradermal delivery to intramuscular delivery, we assumed that a difference of ≥ 15% in seroconversion is clinically significant. To detect this difference in seroconversion with 80% power and a 2-sided 5% significance level, we need 87 participant-students in each group (PASS 2023 software, NCSS, LLC, Kaysville, Utah, USA). With three arms (intramuscular, aerosol, and intradermal), and allowing for 10% dropout, we aim to recruit 100 students per group (300 students in total). We anticipate that we can achieve this recruitment from our target cohort over two years and will make appropriate adjustments to study sites and recruitment strategy if the review of year 1 results indicates this is necessary.

**Statistical analysis.** The reporting of baseline participant characteristics (e.g., demographic profile) and RIVM serologic assays (pre-and post-vaccination) will be by frequency and percentages for categorical variables and mean and standard deviation for continuous variables. The geometric mean concentrations (GMC) for antibody levels will be reported in mean and 95% confidence intervals. The primary analysis will determine the proportion of participants (in frequency and percentages) achieving seropositivity (i.e., protective antibody titers, ≥ 120mIU/ml) and 2- and 4-fold seroconversions from baseline at day 28 and 12-months post-intervention in each intervention group. The pre- to

post-vaccination antibody level changes for each vaccination method will also be displayed using reverse cumulative distribution plots. The frequency and percentages of reported side effects by intervention arms will be calculated and differences between arms compared using the chi-square test. The association of 2- and 4-fold seroconversion with other variables will be determined using univariate and multivariate logistic regression models. The predictors of post-vaccination GMC changes will be estimated using linear regression models. For the regression models, we will determine a predictor-outcome association for the following covariates: sex, age, ethnicity, country of birth, smoking, alcohol intake, history of allergy, medication use, co-morbidities, co-administration of any other vaccine, intervention arm allocation, and baseline antibody titer. The frequency and percentages of participants with detectable measles vaccine virus in oral secretions will be reported and cross-tabulated against 2- and 4-fold antibody titer rise from baseline at 28 days and 12 months post-vaccination follow-up. Once results from participants vaccinated in 2023 are available, an interim analysis will be performed to examine serologic responses and adverse effects among participants by study arm.

Participant attrition is anticipated due to the prolonged trial duration and the multiple study visit requirements for biological sample donation. Therefore, the analysis will only include the data available at different time points (per protocol analysis). The missing data will be reported in frequency and percentages and discussed narratively.

All analyses will be performed using the Stata statistical software version 18.0 (StataCorp, College Station, Texas, USA), and the statistical significance will be determined by $p < 0.05$.

## Protocol amendments

While the trialists aim to follow the protocol strictly, amendments may be required to address any unforeseen challenges that might arise. For instance, if most participants fail to attain protective antibody titers with an alternative vaccination method (e.g., aerosol), the research team may plan to revaccinate them with a different vaccination method after obtaining clearance from relevant regulatory bodies and ethics committees.

## Confidentiality

Participant identity will remain concealed at all times. Only de-identified datasets will be used for the analyses. The destruction of biological samples sent to RIVM will occur five years after receipt unless specific consent is obtained for longer retention. The research team (the principal investigator and designated individuals) will retain participant data (in de-identified form) in secure locations (web servers) for ten years following project completion.

## Dissemination of trial results

Dissemination of trial results will be through presentations, conferences, and publication. We will report trial findings irrespective of their statistical significance. We will provide feedback as lay summaries to participants and community representatives, including Māori participants; a hui process with Māori and Pacific people will also be used given the opportunities to optimize vaccine uptake and outcomes.[40] Hui is an important ritual in Māori communities, means organizing meetings or gatherings to generate knowledge by collecting, generating, and sharing information and is part of the methodology used in this study.[41, 42]

## Name and contact information for the trial sponsor

Prof Peter McIntyre, Department of Paediatrics and Child Health, University of Otago, New Zealand

## Supporting information

**S1 File. SPIRIT 2013 Checklist.**
(PDF)

**S2 File. Forms used to capture participant information at different time points.**
(PDF)

**S3 File. Written consent form.**
(PDF)

**S4 File. Diagram of pulmonary and intradermal vaccine delivery tools used in this trial.**
(PDF)

**S1 Protocol. Clinical Trial of Alternate Routes of Measles-Mumps-Rubella vaccine administration in vaccinated young adults.**
(PDF)

## Author contributions

**Conceptualization:** Sumanta Saha, Melanie Millier, Ari Samaranayaka, Liza Edmonds, Emma Best, James Ussher, Andrew Anglemyer, Jennifer Lee, Michael Tatley, Felicity Cutts, Rob van Binnendijk, Peter McIntyre.

**Funding acquisition:** Sumanta Saha, Peter McIntyre.

**Methodology:** Sumanta Saha, Melanie Millier, Ari Samaranayaka, Liza Edmonds, Emma Best, James Ussher, Andrew Anglemyer, Jennifer Lee, Michael Tatley, Felicity Cutts, Rob van Binnendijk, Peter McIntyre.

**Supervision:** Ari Samaranayaka, Peter McIntyre.

**Writing – original draft:** Sumanta Saha.

**Writing – review & editing:** Sumanta Saha, Melanie Millier, Ari Samaranayaka, Liza Edmonds, Emma Best, James Ussher, Andrew Anglemyer, Jennifer Lee, Michael Tatley, Felicity Cutts, Rob van Binnendijk, Peter McIntyre.

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
