## [Decision Letter · Decision Letter 0]

16 Oct 2024

PONE-D-24-12770Immunogenicity and safety of measles-mumps-rubella vaccine delivered by the aerosol, intradermal and intramuscular routes in previously vaccinated young adults:  a randomized controlled trial protocolPLOS ONE

Dear Dr. Saha,

Thank you for submitting your manuscript to PLOS ONE. After careful consideration, we feel that it has merit but does not fully meet PLOS ONE’s publication criteria as it currently stands. Therefore, we invite you to submit a revised version of the manuscript that addresses the points raised during the review process.

We look forward to receiving your revised manuscript.

Kind regards,

Arnaud John Kombe Kombe, PhD

Guest Editor

PLOS ONE

Journal Requirements:

1. When submitting your revision, we need you to address these additional requirements. Please ensure that your manuscript meets PLOS ONE's style requirements, including those for file naming. The PLOS ONE style templates can be found at https://journals.plos.org/plosone/s/file?id=wjVg/PLOSOne_formatting_sample_main_body.pdf and https://journals.plos.org/plosone/s/file?id=ba62/PLOSOne_formatting_sample_title_authors_affiliations.pdf 2. If any supporting files for review show as item type ‘other’ please change to item type ‘supporting info’ as the reviewer does not have access to these ’other’ files. 3. We note that the grant information you provided in the ‘Funding Information’ and ‘Financial Disclosure’ sections do not match.  When you resubmit, please ensure that you provide the correct grant numbers for the awards you received for your study in the ‘Funding Information’ section. 4. Thank you for stating the following financial disclosure: "Professor Peter McIntyre has received funding support from Health Research Council, New Zealand for the trial (https://www.hrc.govt.nz/resources/research-repository/measuring-and-boosting-waning-immunity-measles-young-adults). Dr Sumanta Saha received funding for working on this project from Ministry of Business, Innovation and Employment (Funding reference no.TN/PLSP23/46/OUPM; via Te Niwha)." Please state what role the funders took in the study.  If the funders had no role, please state: "The funders had no role in study design, data collection and analysis, decision to publish, or preparation of the manuscript." If this statement is not correct you must amend it as needed. Please include this amended Role of Funder statement in your cover letter; we will change the online submission form on your behalf. 5. Your ethics statement should only appear in the Methods section of your manuscript. If your ethics statement is written in any section besides the Methods, please move it to the Methods section and delete it from any other section. Please ensure that your ethics statement is included in your manuscript, as the ethics statement entered into the online submission form will not be published alongside your manuscript.

Reviewers' comments:

Reviewer's Responses to Questions

**Comments to the Author**

1. Does the manuscript provide a valid rationale for the proposed study, with clearly identified and justified research questions?

Reviewer #1: Partly

Reviewer #2: Yes

2. Is the protocol technically sound and planned in a manner that will lead to a meaningful outcome and allow testing the stated hypotheses?

Reviewer #1: Partly

Reviewer #2: Yes

3. Is the methodology feasible and described in sufficient detail to allow the work to be replicable?

Reviewer #1: Yes

Reviewer #2: Yes

4. Have the authors described where all data underlying the findings will be made available when the study is complete?

Reviewer #1: Yes

Reviewer #2: Yes

5. Is the manuscript presented in an intelligible fashion and written in standard English?

Reviewer #1: Yes

Reviewer #2: Yes

6. Review Comments to the Author

You may also provide optional suggestions and comments to authors that they might find helpful in planning their study.

Reviewer #1: The manuscript could be further improved based on the comments below:

The following require revision:

i) Line 177-178, 191-192

ii) Page 10. The sentence ‘Redness and swelling are graded as absent, mild (grade 1), moderate (grade 2), or severe (grade 3) based on characteristics categorized as >2-5 cm (mild), >5-10 cm (moderate), and >10 cm (severe).’

iii) Page 11, the sentence ‘principal investigator or the sponsor or designee, it results in outcomes’

Page 10, state if $125 in NZD or USD currency.

Page 10, typo 'laboratory (RIVM) The principal’

Page 11, 2023-24 to be written as 2023-2024

For sample size calculation, detailed information, calculation, the comparison groups for the outcome variables are to be provided. Among the three comparison groups (intramuscular, aerosol, and intradermal), which comparison sample size was used for representation and state whether formula or software/sample size calculator was used to derive the sample size.

Page 11, the section Analysis is to be stated as Statistical Analyses.

Page 11, concentrations (GMC) for antibody levels are usually presented as geometric mean with the geometric standard deviation (GSD) or with confidence intervals.

Page 12, ‘chi-square test statistic’ to be written as ‘chi-square test’.

Page 12, the sentence ‘changes from baseline at these two post-vaccination time points…’ requires revision.

Page 12, the sentence ‘The following variables will be considered as candidate covariates in models:’ to be written as ‘The following variables will be considered as candidate covariates in the models’

Page 12, the sentence ‘While we aim to perform the above analyses in an intention-to-treat manner; given the trial's long-term multiple post-intervention follow-up requirements, participant attrition is possible. Therefore, the primary analysis will follow a per-protocol assessment.’ is unclear and requires revision.

Subtitle interim analysis, analysis population, addressing missing data and analysis software are to be omitted and the content is to be combined.

The one or two-tailed p-value is to be stated.

Page 13, for the statement 'in secure locations for ten years’’ states the location e.g. server etc.

The list of references did not conform to the journal’s format.

Supplementary files S1-S5 were not available in the PDF version.

Reviewer #2: The study protocol titled <<immunogenicity aerosol="" and="" by="" delivered="" measles-mumps-rubella="" of="" safety="" the="" vaccine="">> proposed by Sumanta Saha et al., aims at comparing aerosol and intradermal delivery of MMR vaccine with the traditional intramuscular route in seronegative adults in an open label parallel three-arm randomized-controlled trial, particularly its immunogenicity and safety. The scientific question raised by the authors is of great importance as it highlights the failure of the second dose of measles vaccine in providing protection in young adults and adults and the necessity of introducing a third one. Furthermore, the hypotheses are sustainable and the methodology proposed is feasible. The ethics is provided. However, to be considered for publication in PLOS ONE Journal, the authors might consider the following:

Minor revisions

1) The authors may reconsider the following sentence << Only necessary data for specimen identification will be sent to the testing laboratory (RIVM) The principal investigator and designated members will monitor data entry at three monthly intervals and decide on any amendments needed (safety monitoring stated below)>> as follows:

a) << Only necessary data for specimen identification will be sent to the testing laboratory (RIVM). The principal investigator and designated members will monitor data entry at three monthly intervals and decide on any amendments needed (safety monitoring stated below)>>,

b) or << Only necessary data for specimen identification will be sent to the testing laboratory (RIVM), and the principal investigator and designated members will monitor data entry at three monthly intervals and decide on any amendments needed (safety monitoring stated below)>>

2) The authors should mention injection sites for intramuscular and intradermal delivery.

3) Please check, is it <<analysis population="">> or <<population analysis="">> after the <<interim analysis="">> section?

4) In the following sentence, is it <<it’s>> or <<its>>: <<we acknowledge="" niwha="" te="">>.

5) At the beginning of the ‘’Introduction section’’ (L47-64), could the authors give more information about measles causative agent (definition, mode of replication, and transmission), the manifestation of the disease (clinical symptoms) and provide data of previous measles outbreaks following secondary-dose failure?

6) Receipt of ‘’two doses of MMR vaccine’’ and have ‘’measles IgG antibody levels which are negative or equivocal on commercial assays’’ are two important inclusion criteria for this study protocol. We suggest that participants without any proof of written records be excluded from the study (L15).

7) The study will compare aerosol and intradermal delivery of MMR vaccine with the traditional intramuscular route. The authors only described aerosol and intradermal delivery procedures in <<intervention section="">> but omitted to provide details as well for intramuscular delivery although the latter is a traditional route.

8) L192 Are the specific training videos for intervention providers already made?

9) L205 Why the oral fluid collection will only be made during the first three follow-up visits within two weeks post-vaccination and not until follow-up 5 like venous blood collection (L195)?

10) L211-212 <<oral be="" by="" chain="" fluid="" polymerase="" reaction="" reverse="" specimens="" tested="" transcription="" will="">>. Did the authors plan to add another technique to confirm their RT-PCR results? If yes, would the authors like to describe the latter?

11) Would the authors like to specify whether or not there is a control group participant (no vaccine administration)?</oral></intervention></we></its></it’s></interim></population></analysis></immunogenicity>

7. PLOS authors have the option to publish the peer review history of their article (what does this mean? ). If published, this will include your full peer review and any attached files.

**Do you want your identity to be public for this peer review?** For information about this choice, including consent withdrawal, please see our Privacy Policy .

Reviewer #1: No

Reviewer #2: **Yes: ** Ulrich Aymard Ekomi Moure

---

## [Author Response · Author response to Decision Letter 0]

26 Nov 2024

Dear Editors,

We thank you for the thorough review of our manuscript and detailed feedback. Below, are the pointwise responses to your comments and amended a few sentences to ensure contextuality and flow. Along with this document, we have uploaded a clean manuscript and a version with tracked changes (in colored text). These responses are also uploaded in a tabulated form as a separate document.

Thank you.

REVIEWER 1 COMMENTS

1. The following require revision:177-178

Authors’ reply: Following was the sentence in line 177-178: ‘As vaccine potency can vary depending on lot,[33] we will make every attempt to use the same lot unless faced with situations that are beyond the control of the research team, such as supply shortage, vaccine vials crossing expiry date before vaccination day, damage of vaccine potency (e.g., cold chain failure), etc.’

We have updated the above sentence as below-

‘Though using the same vaccine lot would be ideal, it would not be feasible as our study will be conducted at different geographic locations over a prolonged period (up to 3 years). However, the vaccine lots will be recorded, and antibody responses by vaccine lot will be examined.’

2. The following require revision: 191-192

Authors’ reply: Following was the sentence in line 191-192: ‘All vaccinators will hold a current authorized vaccinator certificate and will prepare and administer intradermal and aerosol vaccinations utilizing specific training videos and manufacturer instructions.’

We have updated the above sentence as below:

‘All trial participants will be vaccinated by trained vaccinators equipped for vaccination by the intradermal and aerosol route through familiarity with manufacturer instructions and training videos for the specific devices used.’

3. The following require revision: Page 10. The sentence ‘Redness and swelling are graded as absent, mild (grade 1), moderate (grade 2), or severe (grade 3) based on characteristics categorized as >2-5 cm (mild), >5-10 cm (moderate), and >10 cm (severe).’

Authors’ reply: The revised sentence is below:

‘Redness and swelling are graded as absent, mild (grade 1; >2-5 cm), moderate (grade 2; >5-10 cm), or severe (grade 3; >10 cm).’

4. The following require revision: Page 11, the sentence ‘principal investigator or the sponsor or designee, it results in outcomes’

Authors’ reply: The revised sentence is below:

‘The principal investigator or the sponsor or designee will categorize a side effect as serious if it leads to one of the following: a life-threatening event that puts a participant at immediate risk of death and hospitalization.’

5. Page 10, state if $125 in NZD or USD currency.

Authors’ reply: We have clarified in the manuscript that payments are in New Zealand dollars.

6. Page 10, typo 'laboratory (RIVM) The principal’

Authors’ reply: Corrected.

7. Page 11, 2023-24 to be written as 2023-2024

Authors’ reply: Corrected.

8. For sample size calculation, detailed information, calculation, the comparison groups for the outcome variables are to be provided. Among the three comparison groups (intramuscular, aerosol, and intradermal), which comparison sample size was used for representation and state whether formula or software/sample size calculator was used to derive the sample size.

Authors’ reply: Following sample size calculation description is from the submitted version:

‘Over the two years 2023-24, we anticipate ~1400 students will be screened for eligibility. To compare vaccine delivery methods, we assumed that a difference of ≥15% in seroconversion is clinically significant. To detect this difference in seroconversion with 80% power and a 5% significance level, we need 87 participant-students with 1:1 allocation ratio. With three arms (intramuscular, aerosol, and intradermal), and allowing for 10% dropout, we have a target of 300 students. We anticipate that we can achieve our recruitment target of 300 participants (100 per arm) in our target cohort over two years and will make appropriate adjustments to study sites and recruitment strategy if the review of year 1 results indicates this is necessary.’

We amended the description of sample size estimation as below to include requested additional information:

‘Over the two years 2023-2024, we anticipate ~1400 students will be screened for eligibility. When comparing efficacy of aerosol delivery and intradermal delivery to intramuscular delivery, we assumed that a difference of ≥15% in seroconversion is clinically significant. To detect this difference in seroconversion with 80% power and a 2-sided 5% significance level, we need 87 participant-students in each group (PASS 2023 software, NCSS, LLC, Kaysville, Utah, USA). With three arms (intramuscular, aerosol, and intradermal), and allowing for 10% dropout, we aim to recruit 100 students per group (300 students in total). We anticipate that we can achieve this recruitment from our target cohort over two years and will make appropriate adjustments to study sites and recruitment strategy if the review of year 1 results indicates this is necessary’

9. Page 11, the section Analysis is to be stated as Statistical Analyses.

Authors’ reply: We updated it in the manuscript.

10. Page 11, concentrations (GMC) for antibody levels are usually presented as geometric mean with the geometric standard deviation (GSD) or with confidence intervals.

Authors’ reply: Thanks for pointing it out. Although we mentioned how we will analyze the continuous variables in general, we added the following sentence under the ‘Statistical analysis’ section with respect to GMC to clarify it:

‘The geometric mean concentrations (GMC) for antibody levels will be reported in mean and 95% confidence intervals.’

11. Page 12, ‘chi-square test statistic’ to be written as ‘chi-square test’.

Authors’ reply: We updated it in the manuscript.

12. Page 12, the sentence ‘changes from baseline at these two post-vaccination time points...’ requires revision.

Authors’ reply: The sentence in the submitted version was: ‘The determinants of GMC changes from baseline at these two post-vaccination time points will be determined using univariate and multivariate linear regression models.’

The revised sentence is below:

‘The predictors of post-vaccination GMC changes will be estimated using linear regression models.’

13. Page 12, the sentence ‘The following variables will be considered as candidate covariates in models:’ to be written as ‘The following variables will be considered as candidate covariates in the models’

Authors’ reply: The revised sentence is below:

‘For the regression models, we will determine a predictor-outcome association for the following covariates: sex, age, ethnicity, country of birth, smoking, alcohol intake, history of allergy, medication use, co-morbidities, co-administration of any other vaccine, intervention arm allocation, and baseline antibody titer.’

14. Page 12, the sentence ‘While we aim to perform the above analyses in an intention-to-treat manner; given the trial's long-term multiple post-intervention follow-up requirements, participant attrition is possible. Therefore, the primary analysis will follow a per-protocol assessment.’ is unclear and requires revision.

Authors’ reply: The revised sentence is below:

‘Participant attrition is anticipated due to the prolonged trial duration and the multiple study visit requirements for biological sample donation. Therefore, the analysis will only include the data available at different time points (per protocol analysis). The missing data will be reported in frequency and percentages and discussed narratively.’

15. Subtitle interim analysis, analysis population, addressing missing data and analysis software are to be omitted and the content is to be combined.

Authors’ reply: We have omitted the subtitles and combined the content.

16. The one or two-tailed p-value is to be stated.

Authors’ reply: We stated it in the sample size description (mentioned above; Reviewer 1, comment 8).

17. Page 13, for the statement 'in secure locations for ten years’’ states the location e.g. server etc.

Authors’ reply: We will be keeping the data in secure web servers. The revised sentence is below:

‘The research team (the principal investigator and designated individuals) will retain participant data (in de-identified form) in secure locations (web servers) for ten years following project completion.’

18. The list of references did not conform to the journal’s format.

Authors’ reply: We have updated the formatting of the references following the journal guidelines.

19. Supplementary files S1-S5 were not available in the PDF version.

Authors’ reply: While uploading the revised version, we have uploaded supplementary documents in pdf version. If there is any difficulty in accessing these documents, please let us know.

REVIEWER 2 COMMENTS

20. The authors may reconsider the following sentence << Only necessary data for specimen identification will be sent to the testing laboratory (RIVM) The principal investigator and designated members will monitor data entry at three monthly intervals and decide on any amendments needed (safety monitoring stated below)>> as follows:

a) << Only necessary data for specimen identification will be sent to the testing laboratory (RIVM). The principal investigator and designated members will monitor data entry at three monthly intervals and decide on any amendments needed (safety monitoring stated below)>>,

b) or << Only necessary data for specimen identification will be sent to the testing laboratory (RIVM), and the principal investigator and designated members will monitor data entry at three monthly intervals and decide on any amendments needed (safety monitoring stated below)>>

Authors’ reply: Thanks for the suggestions. We opted for the second edit you suggested and updated it in the revised manuscript.

21. The authors should mention injection sites for intramuscular and intradermal delivery.

Authors’ reply: Thanks for letting us know. Given the frequent use of intramuscular routes to inject various intramuscular vaccines (e.g., MMR, hepatitis B, Tetanus-Diphtheria vaccine), we assumed it common knowledge and didn’t mention it. However, following your advice, we have briefly mentioned the intramuscular vaccine delivery site in the following manner:

‘After reconstituting the vaccine in 0.5 ml of diluent, the vaccine will be injected at a 90-degree angle in the outer part of the upper arm using a 23–25-gauge needle to deliver the vaccine deep into the deltoid muscle.’

Regarding the intradermal vaccine delivery, we have already mentioned the vaccine delivery site. For your reference, we are copying it below from the manuscript:

‘Intradermal administration of vaccine uses the Micronjet600TM device (NanoPass Technologies Ltd, Israel), a device consisting of three 600 µm long microneedles (https://www.nanopass.com/). PRIORIX MMR vaccine is reconstituted to a total volume of 0.3 mL of sterile water and drawn up into a 2mL syringe. After attachment to the syringe, the microneedle is inserted into the participant’s skin in the deltoid region at a 45-degree angle in accordance with manufacturer instructions, and the vaccine dose is manually delivered to the intradermal region. A visible bleb indicates successful intradermal delivery.’

22. At the beginning of the ‘’Introduction section’’ (L47-64), could the authors give more information about measles causative agent (definition, mode of replication, and transmission), the manifestation of the disease (clinical symptoms) and provide data of previous measles outbreaks following secondary-dose failure?

Authors’ reply: We included the following sentences to mention measles definition, mode of replication, transmission, and clinical symptoms:

‘Measles is a highly contagious infectious disease caused by the measles virus, a negative-sense single-stranded RNA virus of the Morbillivirus genus and Paramyxoviridae family.(1) It has an incubation period of 10-14 days. On entering the epithelial cells of the upper airway of a susceptible host, the virus first infects the myeloid cells (e.g., macrophages and dendritic cells),(2,3) then it migrates to infect regional lymphoid cells and infects T- and B-cells, which eventually enter the bloodstream and infect the epithelial cells of the skin and submucosa of the respiratory tract. The latter results in coryzal symptoms and cough with spread of infectious viral particles via respiratory droplets and small aerosols to other individuals.(4,5) Measles causes a severe systemic disease clinically characterized by symptoms including fever, cough, coryza, conjunctivitis, and rash, but in high income settings rarely mortality.(1)’

We included the following sentences to report data on measles occurrence in two doses of measles containing vaccine recipients during previous outbreaks has been added as below:

‘As measles remain endemic in different parts of the globe, individual countries where elimination was achieved following prolonged periods of high two-dose measles vaccination coverage and very low measles incidence have increasingly documented measles infection in people with two documented measles vaccine doses. For example, 26/232 (11%) cases in US outbreaks (California; 2000-2015),(8) 6/32 (19%) cases in an Australian outbreak (New South Wales 2012),(9) and 11/94 (11.7%) cases during a South Korean outbreak in 2019.(10)’

23. Receipt of ‘’two doses of MMR vaccine’’ and have ‘’measles IgG antibody levels which are negative or equivocal on commercial assays’’ are two important inclusion criteria for this study protocol. We suggest that participants without any proof of written records be excluded from the study (L15).

Authors’ reply: Much appreciated suggestion. However, we would respectfully like to keep participants who signed a declaration in the trial due to the reasons stated below:

In the absence of a participant’s vaccination record, an ideal way of verifying the receipt of second dose of measles vaccine would be from a national immunization registry (after obtaining required ethical approval); however, historically, New Zealand didn’t have such national register of childhood immunizations until 2005. Since we are recruiting individuals aged ≥18 (as per the eligibility criteria), the youngest enrolled in the first trial year (2023) would have been born on or before 2005. This means that verifying self-declared receipt of 2nd dose of MMR will not be possible from written records. In light of this, we decided to accept individuals who had signed a declaration to vouch for their childhood receipt of two doses of MMR vaccine in the absence of written records as eligible. This is one of the limitations of our trial, which we will acknowledge it while reporting our trial findings and also examine results among those with and without written records.

24. The study will compare aerosol and intradermal delivery of MMR vaccine with the traditional intramuscular route. The authors only described aerosol and intradermal delivery procedures in <> but omitted to provide details as well for intramuscular delivery although the latter is a traditional route.

Authors’ reply: Intramuscular delivery is now included in the revised manuscript.

25. L192 Are the specific training videos for intervention providers already made?

Authors’ reply: These training videos come from the manufacturers.

The website links to such videos are shared here:

• For aerosol delivery: https://www.aerogen.com/products/aerogen-solo

• For intradermal delivery: https://www.nanopass.com/micronjet-microneedle-device/instructions-for-use/

26. L205 Why the oral fluid collection will only be made during the first three follow-up visits within two weeks post- vaccination and not until follow-up 5 like venous blood collection (L195)?

Authors’ reply: The purpose of oral fluid collection in the study is to examine whether measles vaccine virus (MVV) can be detected – which we assume would indicate that sterilizing immunity to measles infection was not present. There are limited data about the prevalence of detection of MVV post vaccination and the data which are available relate to children receiving the vaccine for the

---

## [Decision Letter · Decision Letter 1]

10 Dec 2024

PONE-D-24-12770R1Immunogenicity and safety of measles-mumps-rubella vaccine delivered by the aerosol, intradermal and intramuscular routes in previously vaccinated young adults: a randomized controlled trial protocolPLOS ONE Dear Dr. Saha,

Thank you for submitting the revised version of your manuscript to PlosOne. We appreciate the effort you have invested in addressing the reviewers’ comments and improving the quality of your work.

After carefully reviewing your revised manuscript, we find that it has been significantly improved and aligns well with many of the reviewers' initial recommendations. However, to ensure the highest quality and suitability for publication, additional revisions are required before we can proceed further.

The detailed feedback from the reviewers (Reviewer 2), along with specific points for revision, are bellow for your reference. We encourage you to carefully consider these suggestions and make the necessary modifications to strengthen your manuscript.

We look forward to receiving your revised manuscript.

Kind regards,

Arnaud John Kombe Kombe, PhD

Guest Editor

PLOS ONE

Journal Requirements:

Reviewers' comments:

Reviewer's Responses to Questions

**Comments to the Author**

1. Does the manuscript provide a valid rationale for the proposed study, with clearly identified and justified research questions?

Reviewer #1: Yes

Reviewer #2: Yes

2. Is the protocol technically sound and planned in a manner that will lead to a meaningful outcome and allow testing the stated hypotheses?

Reviewer #1: Yes

Reviewer #2: Yes

3. Is the methodology feasible and described in sufficient detail to allow the work to be replicable?

Reviewer #1: Yes

Reviewer #2: Yes

4. Have the authors described where all data underlying the findings will be made available when the study is complete?

Reviewer #1: No

Reviewer #2: Yes

5. Is the manuscript presented in an intelligible fashion and written in standard English?

Reviewer #1: Yes

Reviewer #2: Yes

6. Review Comments to the Author

You may also provide optional suggestions and comments to authors that they might find helpful in planning their study.

Reviewer #1: The authors have adequately addressed the comments. I have no further comments. The manuscript is suitable for publication.

Reviewer #2: After a careful review of the modified manuscript version submitted by the authors, the major issues have been well-addressed and robust arguments have been provided where more light where needed. Besides, the revised manuscript submitted is presented in an intelligible fashion. However, the English throughout the manuscript needs further corrections. Based on that, I suggest for this study protocol to be eligible for publication by PLOS ONE Journal, the necessity of addressing the following minor issues:

1. L21 the word ‘re-introduction’ is not appropriate.

2. Check the use or not of ‘the’ before ‘measles vaccine’ in L29, L31, and L54, respectively and all over the manuscript.

3. Check the plural form in L25 ‘administration’, L96-97 ‘aerosol and intradermal delivery’, L107 ‘delivery’, and L108 and L125 ‘antibody’.

4. L51-52, L94, and in the entire manuscript, what do the authors mean by ‘settings’?

5. This sentence needs to be revised: ‘As measles remain endemic in different parts of the globe, individual countries where elimination was achieved following prolonged periods of high two-dose measles vaccination coverage and very low measles incidence have increasingly documented measles infection in people with two documented measles vaccine doses. For example, 26/232 (11%) cases in US outbreaks (California; 2000-2015), (8) 6/32 (19%) cases in an Australian outbreak (New South Wales 2012), (9) and 11/94 (11.7%) cases during a South Korean outbreak in 2019. (10)’

6. Similarly, add data related to previous measles outbreaks in New Zealand. As the study is being conducted in New Zealand, this would give an overview of past measles infections in that country and the necessity to conduct the current study. The authors can refer to S2 File, especially in the Section ‘Importance’.

7. L84-89 This sentence needs to be re-checked.

8. L60 ‘As measles remain’, L94 ‘measles is’, and in the entire manuscript: in what context(s) this word is used so that the authors use plural or singular forms after it?

9. L112-113, write entirely ‘December’ like you wrote entirely ‘February’, and not ‘Dec’.

10. The use or not of ‘the’ before some nouns/group of nouns should be checked in the entire manuscript.

7. PLOS authors have the option to publish the peer review history of their article (what does this mean? ). If published, this will include your full peer review and any attached files.

**Do you want your identity to be public for this peer review?** For information about this choice, including consent withdrawal, please see our Privacy Policy .

Reviewer #1: No

Reviewer #2: No

---

## [Author Response · Author response to Decision Letter 1]

22 Dec 2024

Dear Reviewers,

Thank you for your feedback. Based on your comments, we have amended the manuscript further and tweaked a few sentences to enhance clarity. Along with this document, we have uploaded a clean manuscript and a version with tracked changes (in colored text). Please find below the pointwise responses to your comments (the same can be found in a tabulated form in the attached file named ‘Response to Reviewers.’

Thank you.

Reviewers’ comment

Reviewer #1:

The authors have adequately addressed the comments. I have no further comments. The manuscript is suitable for publication.

Authors’ reply

Thank you for your feedback.

Reviewer #2:

1. L21 the word ‘re-introduction’ is not appropriate.

Authors’ reply

Done.

The revised sentence reads as the following-

‘There are increasing reports of outbreaks of measles in countries that had eliminated the infection using two doses of measles-mumps-rubella (MMR) vaccine.’

2. Check the use or not of ‘the’ before ‘measles vaccine’ in L29, L31, and L54, respectively and all over the manuscript.

Authors’ reply

Done.

We checked and updated it in a few instances where it appeared appropriate.

3. Check the plural form in L25 ‘administration’, L96-97 ‘aerosol and intradermal delivery’, L107 ‘delivery’, and L108 and L125 ‘antibody’. Done.

Authors’ reply

We checked and updated the plural forms as advised.

4. L51-52, L94, and in the entire manuscript, what do the authors mean by ‘settings’?

Authors’ reply

Settings is a general word commonly used by the World Health Organization (WHO) and other authors to designate situations and places, including countries.

E.g.,

1. Bolotin, Shelly et al. “In Elimination Settings, Measles Antibodies Wane After Vaccination but Not After Infection: A Systematic Review and Meta-Analysis.” The Journal of infectious diseases vol. 226,7 (2022): 1127-1139. doi:10.1093/infdis/jiac039

2. Measles outbreaks strategic response plan 2021–2023. Geneva: World Health Organization; 2021. (https://www.who.int/publications/i/item/9789240018600)

5. This sentence needs to be revised: ‘As measles remain endemic in different parts of the globe, individual countries where elimination was achieved following prolonged periods of high two-dose measles vaccination coverage and very low measles incidence have increasingly documented measles infection in people with two documented measles vaccine doses. For example, 26/232 (11%) cases in US outbreaks (California; 2000-2015), (8) 6/32 (19%) cases in an Australian outbreak (New South Wales 2012), (9) and 11/94 (11.7%) cases during a South Korean outbreak in 2019. (10)’

Authors’ reply

Done.

The updated sentence reads as the following:

‘Measles remains endemic in many countries. As a result, countries where elimination has been achieved following prolonged periods of high two-dose measles vaccination coverage accompanied by very low measles incidence, when outbreaks occur due to imported cases, have increasingly documented measles infection in people with two documented measles vaccine doses. Examples include US outbreaks where 26/232 (11%) cases in California (2000-2015),[8] 6/32 (19%) cases in an Australian outbreak (New South Wales 2012),[9] and 11/94 (11.7%) cases during a South Korean outbreak in 2019.[10]’

6. Similarly, add data related to previous measles outbreaks in New Zealand. As the study is being conducted in New Zealand, this would give an overview of past measles infections in that country and the necessity to conduct the current study. The authors can refer to S2 File, especially in the Section ‘Importance’.

Authors’ reply

Done.

The following sentence has been added to address it:

‘Similarly, in a large outbreak of over 2100 cases in New Zealand in 2019, about 10% of cases in young adults occurred in two-dose MMR recipients.[11]’

Please note that reference 11 is a new one added to the manuscript.

Reference:

[11] Institute of Environmental Science and Research. Measles weekly reports - 2020, https://www.esr.cri.nz/digital-library/measles-weekly-reports-2020/ (2020, accessed 21 December 2024).

7. L84-89 This sentence needs to be re-checked. Done.

Authors’ reply

The updated sentence reads as the following:

‘VMNs generate 3-5 µm size particles that deposit the vaccine deep into the lung tissue when inhaled.[19] Other than the study cited above,[16] previous studies in adults on the safety and immunogenicity of measles-containing vaccines (MCV) by route of administration have compared subcutaneous administration with aerosol delivery using the Classical Mexican Device for aerosol generation, used vaccine strains not used in high-income elimination settings (such as Edmonston Zagreb (measles) and Leningrad Zagreb or Rubini (mumps strains)), and have predominantly included seropositive subjects.[20, 21]’

8. L60 ‘As measles remain’, L94 ‘measles is’, and in the entire manuscript: in what context(s) this word is used so that the authors use plural or singular forms after it?

Authors’ reply

L60 has been amended as below:

‘Measles remains endemic in many countries. As a result, countries where elimination has been achieved following prolonged periods of high two-dose measles vaccination coverage accompanied by very low measles incidence, when outbreaks occur due to imported cases, have increasingly documented measles infection in people with two documented measles vaccine doses.’

L94 has been amended as below:

‘Given evidence that measles cases have been increasingly recognised in measles elimination settings in two-dose vaccinated young adults in childhood, whether the strength and/or duration of protection can be improved is a key question.’

9. L112-113, write entirely ‘December’ like you wrote entirely ‘February’, and not ‘Dec’.

Authors’ reply

Done.

The updated sentence reads as the following:

‘The trial started recruiting in February 2023 and will end in December 2025.’

10. The use or not of ‘the’ before some nouns/group of nouns should be checked in the entire manuscript.

Authors’ reply

Please see reply to comment no. 2.

---

## [Editor Report · Decision Letter 2]

24 Jan 2025

Immunogenicity and safety of measles-mumps-rubella vaccine delivered by the aerosol, intradermal and intramuscular routes in previously vaccinated young adults: a randomized controlled trial protocol

PONE-D-24-12770R2

Dear Dr. Saha,

We’re pleased to inform you that your manuscript has been judged scientifically suitable for publication and will be formally accepted for publication once it meets all outstanding technical requirements.

Kind regards,

Arnaud John Kombe Kombe, PhD

Guest Editor

PLOS ONE

---

## [Editor Report · Acceptance letter]

PONE-D-24-12770R2

PLOS ONE

Dear Dr. Saha,

I'm pleased to inform you that your manuscript has been deemed suitable for publication in PLOS ONE. Congratulations! Your manuscript is now being handed over to our production team.

Kind regards,

on behalf of

Dr. Arnaud John Kombe Kombe

Guest Editor

PLOS ONE